# Structure of Polytetrafluoroethylene Modified by the Combined Action of γ-Radiation and High Temperatures

**DOI:** 10.3390/polym13213678

**Published:** 2021-10-25

**Authors:** Alexander Sergeevich Smolyanskii, Ekaterina Dmitrievna Politova, Ol’ga Alekseevna Koshkina, Mikhail Aleksandrovich Arsentyev, Pavel Prokof’evich Kusch, Lev Vladimirovich Moskvitin, Sergei Vital’evich Slesarenko, Dmitrii Pavlovich Kiryukhin, Leonid Izrailevich Trakhtenberg

**Affiliations:** 1High Energy Chemistry and Radioecology Department, D. Mendeleev University of Chemical Technology of Russia, Miusskaya Ploshchad 125047, Moscow, Russia; levmoskvitin@gmail.com; 2Laboratory of Functional Nanocomposites, N.N. Semenov Federal Research Center for Chemical Physics, Russian Academy of Science, Moscow 119991, Moscow, Russia; politova@nifhi.ru (E.D.P.); olga_koshkina_94@mail.ru (O.A.K.); litrakh@gmail.com (L.I.T.); 3Technology Department, Quantum R LLC, Skolkovo 125319, Moscow, Russia; mihail.arsentyev92@gmail.com (M.A.A.); Ser.slesarenko@yandex.ru (S.V.S.); 4Laboratory of Cryochemistry and Radiation Chemistry, The Institute of Problems of Chemical Physics, Russian Academy of Science, Chernogolovka 142400, Moscow, Russia; KPP@icp.ac.ru (P.P.K.); omega@chgnet.ru (D.P.K.); 5Laboratory of Chemical Kinetics, Chemical Department, Lomonosov Moscow State University, Moscow 119991, Moscow, Russia; 6Department of Chemical Physics, Moscow Institute of Physics and Technology (State University), 9 Institutskii Lane, Dolgoprudny 141700, Moscow, Russia

**Keywords:** polytetrafluoroethylene, ionizing radiation, high temperatures, modification, crystallization, cross-linking, pore, structure

## Abstract

By means of X-ray computed microtomography (XCMT), the existence of a developed porous structure with an average pore diameter of ~3.5 μm and pore content of ~1.1 vol.% has been revealed in unirradiated polytetrafluoroethylene (PTFE). It has been found that the combined action of gamma radiation (absorbed dose per PTFE of ~170 kGy) and high temperatures (327–350 °C) leads to the disappearance of the porous structure and the formation of several large pores with sizes from 30 to 50 μm in the bulk of thermal-radiation modified PTFE (TRM-PTFE). It has been established by X-ray diffraction (XRD) analysis that the thermal-radiation modification of PTFE leads to an increase in the interplanar spacings, the degree of crystallinity and the volume of the unit cell, as well as to a decrease in the size of crystals and the X-ray density of the crystalline phase in comparison with the initial polymer. It is assumed that the previously-established effect of improving the deformation-strength and tribological properties of the TRM-PTFE can be due not only to the radiation cross-linking of polymer chains but also to the disappearance of the pore system and to the ordering of the crystalline phase of PTFE.

## 1. Introduction

High-temperature radiation modification of polytetrafluoroethylene (PTFE) is a promising method for overcoming the disadvantages of this material-such as cold brittleness, porosity, and low radiation resistance [1,2]. It was found [3,4,5,6] that the combined action of various types of ionizing radiation (γ-radiation, electrons of different energies) and temperatures exceeding the melting point of polymer crystallites can improve the tribological and deformation-strength characteristics of PTFE, change its spectral-luminescent properties, and increase the radiation resistance.

A common drawback of cited studies [1,2,3,4,5,6] is the lack of estimates of the parameters characterizing the newly-formed three-dimensional network-such as the density of the cross-links and the molecular weight of the TRM-PTFE between cross-links [7]. The conclusion about the occurrence of radiation-induced cross-linking processes during irradiation of PTFE melt is made mainly on the basis of measurements of the deformation-strength and/or tribological characteristics of the thermal-radiation modified polymer [1,2,3,4,5,6].

The lack of estimates of the density of cross-links, which is a key parameter characterizing the efficiency of radiation cross-linking processes, excludes the comparison of experimental data with the predicted efficiency of the high-temperature irradiation treatment of PTFE [1,2,3,4,5,6]. This may be due to the high chemical inertness of PTFE, which makes it difficult to estimate the density of cross-links in TRM-PTFE using the Charlesby–Pinner equation [7].

At the same time, it is known that traces of the crystalline phase of PTFE can exist in the melt up to a temperature of 400 °C and that the processes of radiolytic gas evolution and depolymerization under γ-irradiation of PTFE can proceed for up to 500 °C [8,9]. Therefore, it can be assumed that the irradiation of PTFE in the melt is accompanied by the occurrence of competing processes of radiation destruction and cross-linking, crystallization, gas evolution, etc.

It has been shown that, under the conditions of simultaneous cross-linking and crystallization processes, the cross-link density can be estimated from the average crystallite size [10]. Then, it is relevant to carry out the X-ray diffraction study of radiation-induced changes in the crystalline phase of the initial and TRM-PTFE [11].

The aim of this study is to establish the nature of the effects of improving the deformation-strength and tribological properties, increasing the radiation resistance of PTFE under the combined action of ionizing radiation and high temperatures by studying the microstructure and crystalline phase of the initial and thermal-radiation modified polymer.

## 2. Material and Methods

### 2.1. Materials

For the manufacture of TRM-PTFE and control samples, PTFE grade F-4 (Soviet State Standard 10007-80) produced by JSC “HaloPolymer Kirovo-Chepetsk” (Kirovo-Chepetsk, Perm’ Region, Russian Federation) was used in the form of cylindrical rods 25 mm in diameter and 250 mm in height, which were obtained by sintering powdered PTFE obtained by emulsion polymerization.

Disks with a diameter of 25 mm and a thickness of 2 mm, which were cut from cylindrical rods of the initial and TRM-PTFE, were used for X-ray diffraction studies. The X-ray computed microtomography study of the microstructure of initial and TRM-PTFE was carried out on cylindrical polymer samples with a diameter of 3 mm and a height of 6 mm.

### 2.2. Radiation Modification and Dosimetry 

The thermal-radiation modification of PTFE blanks by exposure to γ-radiation of the ^60^Co isotope (mean energy 1.25 MeV) was implemented in a radiation-chemical apparatus made of stainless steel and located in the working chamber of the unique scientific installation (USI) “Gammatok” at the Institute of Problems of Chemical Physics, Russian Academy of Science [12]. Irradiation was performed in the nitrogen environment at a temperature from 327 to 350 °C and a dose rate of 3 Gy/s to an absorbed dose of 200 kGy. After radiation treatment, the TRM-PTFE sample was cooled to room temperature in the working chamber of the USI “Gammatok”. Control samples of initial PTFE were irradiated under similar conditions, but at a temperature of 60 °C.

The dose rate and absorbed dose were determined using the Fricke method in accordance with the requirements of Russian State Standard 34155-2017. To take into account the difference between the electron densities of the Fricke dosimetric solution and PTFE, when calculating the dose absorbed by the polymer, a correction factor (0.865) was introduced according to the Soviet State Standard 27602-88. Therefore, the dose absorbed by PTFE under the described irradiation conditions was 200 × 0.865 = 173 kGy. Since the dose rate is determined by the Fricke method with an accuracy of 20%, and the PTFE preform during the irradiation was surrounded by a stainless-steel layer, it can be assumed that the condition of electronic equilibrium during the thermal-radiation modification of PTFE is fulfilled.

### 2.3. X-ray Computed Microtomography Measurements

X-ray computed microtomography investigations of the three-dimensional structure in initial polymer and TRM-PTFE in the submicron range (resolution limit 0.7 μm) were carried out using an FEI HELISCAN MICRO-CT X-ray microtomograph (Thermo Fisher Scientific, Waltham, MA, USA) [13].

As is known [14], in the method of X-ray microtomography, the computer recreates the model of the object under study after its layer-by-layer scanning using a narrow X-ray beam (Figure 1). An X-ray tube, rotating around the sample under study, with the help of a narrow beam of X-rays, shines through (scans) its body at different angles, passing through 360° in a full revolution. By the end of the revolution, signals from all detectors are entered into the computer’s memory, then, using computer processing, a planar image is created-a cross-section of the sample at a predetermined point (Figure 2). In the process of measurements, 1024 images of cross-sections of samples of the initial polymer and TPM-PTFE were obtained with the help of which the reconstruction of the PTFE microstructure was further carried out (Figure 1 and Figure 2) [15,16].

Passing through the sample, X-rays are attenuated according to the density and atomic composition of PTFE and TPM-PTFE. In this case, the regions of the supramolecular structure of the polymers in Figure 2, which attenuate X-ray radiation (crystalline phase, etc.), look light on the planar image of the polymer, and the pore cavities that transmit X-rays look dark (according to the principle of conventional X-ray diffraction [14,15,16]).

Since a detailed analysis of all the scans obtained is a complex computer problem that requires the development of special software, this communication presents images obtained in the central part of the studied polymer samples and the result of the reconstruction of the porous structure of PTFE (Figure 1 and Figure 2).

### 2.4. X-ray Diffraction Pattern Measurements

The X-ray diffraction study of the PTFE and TRM-PTFE crystal structures was carried out at room temperature using a DRON-3M diffractometer (Joint Stock Company “Innovation center “Burevestnik””, St. Petersburg, Leningrad region, Russian Federation). Conditions for recording X-ray spectra: Cu K_α_ radiation with a characteristic wavelength of λ_Cu_ = 1.5406 Å, the scanning interval of diffraction angles 2θ from 5 to 80°, the scanning step from 0.05 to 0.02°, exposure from 1 to 20 s. The degree of crystallinity of the studied PTFE samples was determined from the X-ray diffraction data, as well as by differential scanning calorimetry measurements according to [11].

### 2.5. Density Measurements

The densities of the initial polymer and TRM-PTFE were determined by hydrostatic weighing with an accuracy of 0.001 g/cm^3^. The change in the mass of the sample before and after the test was determined by weighing on an AND microbalance model GR 202 (A&D Company, Ltd., Tokyo, Japan) with a readability of 0.01 mg.

## 3. Results

### 3.1. Observation of the Porous Structure and Its Disappearance after Thermal-Radiation Modification in Polytetrafluoroethylene

As a rule, the production of block PTFE samples is carried out by the method of free sintering of polymer powder, which is carried out as a result of heating a product formed from a PTFE powder located outside the mold, up to 380 °C, followed by cooling in an oven (in accordance with the requirements of the Soviet State Standard 17359-82). 

During sintering, the following (internal) processes are possible: change in the size and shape of pores, crystal growth, reduction and leveling of residual (after pressing) stresses, formation of a liquid phase; spatial redistribution of phases, a decrease in the concentration of defects in crystalline phases, etc. [17]. The main disadvantage of the free sintering method is a significant shrinkage of the block PTFE, which can reach 5–7%. Shrinkage of the material occurs during the cooling stage and is accompanied by the appearance of the so-called “shrinkage porosity” in the PTFE volume [18,19].

Therefore, to successfully achieve the objectives of this study, it is necessary, first of all, to control the PTFE samples for the presence of a porous structure and then to establish the effect of thermal radiation treatment on its change.

Obviously, traditional methods of studying the porous structure-mercury porosimetry, low-temperature sorption of nitrogen, etc. [20]-can be used to study open pores, and are not suitable for analyzing the closed pore system in sintered-block PTFE. Non-destructive measurement of closed pores in PTFE can be carried out by methods of attenuation of ultrasonic waves and ionizing radiation particle fluxes (photons of X-ray and gamma radiation, neutrons) [21,22,23]. The review [24] also provides information on the application of electron microscopy [25], nuclear magnetic resonance [26], and electron paramagnetic resonance spectroscopy [27] to determine closed pores in solids.

The study of porous structures by scattering methods allows us determine the porosity of the material under consideration [21,22,23]. It is convenient to study the size and shape of pores using optical and electron microscopy methods [24,25]. Magnetic resonance methods make it possible to estimate the size distribution by analyzing signals from specially introduced probing additives [26,27].

A distinctive feature of the X-ray computer microtomography method, which does not possess the above concerned methods for determining the porous structure, should be considered the possibility of 3D visualization of the image of the porous structure [28,29]. This makes it possible to establish the characteristics of the porous structure and obtain predictive estimates of various physical properties of the object under study (gas permeability, thermal conductivity etc.) as a result of one measuring through the use of computer analysis methods [30,31].

Previously, the analysis of the closed pore system in PTFE was carried out by the method of hydrostatic weighing or by measuring the overall dimensions of the samples before or after annealing [32,33,34,35]. Attempts to use scanning electron microscopy methods [33,36] to estimate the size of pores and to analyze the spectral shape of the diffraction maximum *hkl* = 100 in X-ray spectra of PTFE by the Rietveld method [37,38] are known also.

Obviously, the use of low-sensitivity measurement methods for assessing the porosity of PTFE by changing the overall dimensions of samples and/or hydrostatic weighing requires statistical processing of the obtained estimates of the porosity of PTFE, which can be done by performing several measurements of the mass-dimensional characteristics of polymer samples before and after annealing.

In the case of electron microscopic studies, it is necessary to prepare sections of PTFE samples taken in several parts of the polymer sample by the microtomation method. Next, it is necessary to analyze the traces of pores on the obtained 2D-images of sections of the PTFE sample. The described procedure is laborious and does not allow for making unambiguous conclusions about the characteristics and size distribution of pores.

The estimates of the porosity of PTFE, obtained as a result of the analysis of the spectral shape of the diffraction maximum *hkl* = 100 in the X-ray spectra of PTFE, cannot be considered reliable, since the spectral shape of the crystallinity peak under consideration is also sensitive to the presence of defects, such as dislocations, in the polymer [39].

It can be concluded that the XCMT method has not previously been used to obtain information on the porous structure of PTFE. Really, it was found from the X-ray computed microtomography measurements that the initial PTFE contains numerous pores, which, in accordance with the above, could arise during sintering of the polymer sample (Figure 1a and Figure 2a) [17,18,19]. The sizes of pores varied from 1 to 12 µm and the size distribution maximum was at 3.5 µm. By integrating the pore size distribution function, the porosity of the initial PTFE sample was estimated to be ~1.1% (Figure 1b).

The subsequent thermal-radiation modification of PTFE leads to an almost complete disappearance of pores (Figure 2b). Several large pores with sizes from 20 to 50 μm can be observed in the sample volume, which indicates the dissolution and/or coalescence of pores in the process of thermal-radiation modification of PTFE. In this case, the gases contained inside the pores are released from the polymer melt into the environment [9].

### 3.2. Characterization of X-ray Diffraction Patterns Registered for Initial and Thermal-Radiation Modified Polytetrafluoroethylene

High-temperature radiation modification has no significant effect on the properties of PTFE X-ray spectrum. Diffraction patterns of the initial polymer and TRM-PTFE in the range of diffraction angles 2θ from 5 to 80° contained 13 and 12 Bragg peaks, respectively, as well as three haloes in the ranges 2θ = 10–25° (I), 25–70° (II) and 72–75° (III) (Table 1, Figure 3 and Figure 4a,b).

The diffraction patterns of the initial and TRM-PTFE were indexed by the method described in [40,41,42], taking into account the reference data (protocol JCPDS No. 47-2217 [43]) and the results previously obtained in [44,45,46]. The procedure for determining the degree of crystallinity *X_c_* of polymers was described earlier [11]. Using the well-known Bragg’ law and Debye–Scherrer equation, the interplanar spacing *d_hkl_* and the average crystal size *L* in the initial polymer and TRM-PTFE were calculated (Table 1 and Table 2) [40].

It is usually considered that PTFE crystals at room temperature belong to the hexagonal system with the unit cell containing 15 CF_2_ groups packed into a spiral with seven turns around the hexagonal axis [40,41,42,43,44,45,46]. Gamma-irradiation of PTFE up to 173 kGy with simultaneous exposure to temperatures in the range of 327–350 °C leads to the shift of almost all diffraction maxima towards smaller diffraction angles (Table 1). The shift changes from 0.042° (diffraction maximum *hkl* = 1015) to 0.311° (*hkl* = 220).

The diffraction maximum *hkl* = 100 dominates in the X-ray spectra of the initial polymer and TRM-PTFE (Figure 3, Table 1). Because of thermal-radiation modification, the amplitude of the diffraction maximum *hkl* = 100 decreases by ~10% and its width increased (the full width at the half-maximum (FWHM) increases by ~0.366°).

It was shown that the diffraction maximum *hkl* = 100 consists of two components, which correspond to PTFE crystals which belong to the hexagonal system, but have different interplanar spacings [11]. By approximating the spectral shape of the diffraction maximum studied with two Gaussians, it was found that the maxima of the components of peak *hkl* = 100 are located at 2θ = 18.027 ± 0.003° and 18.087 ± 0.001° for the initial polymer and at 17.832 ± 0.006° and 18.007 ± 0.001° for TRM-PTFE.

In this case, the shift toward smaller diffraction angles for the first and second components was ~0.195° and ~0.08°, respectively. Consequently, the thermal-radiation treatment leads to an increase in the interplanar spacings *d_hkl_* of both types of PTFE crystals, but this increase turns out to be different in them.

It was shown that reflection in the region 2θ ≈ 18° can be attributed to intermolecular scattering by PTFE macromolecules containing conformational defects [44]. Then, it can be concluded that the shift towards smaller reflection angles in the X-ray spectra of TRM-PTFE is due to a decrease in the concentration of conformational defects caused thermal-radiation modification.

This conclusion is confirmed by the data presented in Table 1: the dominance of equatorial reflections (*l* = 0) in the range of diffraction angles 2θ = 30–70° on the diffraction patterns of the initial polymer and TRM-PTFE can be attributed to disordering along the hexagonal axis of the PTFE unit cell (space group P6mm) [45]. It is known that the crystal structure of PTFE is based on the close packing of spiral polymer molecules along the hexagonal axis [44,45,46]. However, in the course of crystallization, the CF_2_ groups that form the PTFE unit cell can be disordered with respect to the rotation angle around the hexagonal axis and the monomer units of the polymer chains that form the PTFE unit cell can be shifted with respect to each other.

Obviously, γ-irradiation of PTFE in the melt leads to a significant decrease in the concentration of conformational defects in polymer chains and to the ordering of the crystal structure. It should be note that a similar conclusion was made when studying the crystal structure of PTFE irradiated by a proton flux with energy of 4 MeV [47].

It is known that PTFE is characterized by the spherulite crystal structure in the form of expanded chain crystals with very thick lamellas (~1000 Å) [47]. Lamellar surfaces refer to (*h*0*l*) reflection planes, where the angle between the direction of the polymer chain and the normal direction of the lamella varies from 20° to 40° [48]. In the considered range of diffraction angles of the X-ray spectrum of PTFE, the diffraction maxima with *hkl* = 107 and 108 satisfy the requirement for reflection from the lamella surface (Figure 4a,b). The combined action of ionizing radiation and high temperatures results in a decrease in the relative intensity of the diffraction maxima 107 and 108 by ~3% (Table 1, Figure 4a,b).

Within the framework of the three-phase model of an amorphous–crystalline polymer, the structure of PTFE can include crystalline, amorphous, and smectic phases [49]. In polyethylene and in poly-L-lactic acid, the smectic phase is a layer of partially ordered polymer chains around the lamella, which has a thickness of ~40–50 Å [49,50]. Possibly, the smectic phase can also be formed on the walls of lamella, as well as pores in PTFE. Then, a decrease in the intensity of diffraction maxima 107 and 108 in the X-ray spectra of TRM-PTFE (Table 1, Figure 4a,b) may be attributed to a decrease in the content of the smectic phase in the composition of PTFE due to the disappearance of pores [51] and the transformation of lamellae into fibrils [52].

This conclusion is confirmed by the scanning electron microscopy data [53], which indicate that irradiation at temperatures exceeding 200 °C leads to the thickening of lamellae in PTFE and to the formation of a fibrillar structure. As noted above, the consolidation of lamellae will be accompanied by a decrease in the lamella surface area following by a decrease in the intensity of the diffraction maxima *hkl* = 107 and 108 in the X-ray scattering spectrum of PTFE.

In the X-ray scattering spectra of PTFE, the amorphous phase is characterized by a diffuse diffraction maximum in the region of 10–25° [11,44]. Gamma-radiation-induced modification of PTFE at high temperatures leads to an insignificant increase (~5%) in the integral intensity of the diffuse diffraction maximum I, and, consequently, in the content of the amorphous phase in the polymer (Table 3).

The origin of halo II can be attributed to intramolecular scattering from disordered polymer chains with a conformation close to that of PTFE crystals, but not included in the crystalline phase; presumably, these macromolecules are included in the smectic phase of the polymer [44,46]. The spectral shape of halo II can be approximated by two Gaussians (Table 3, Figure 4a,b), which suggests the existence of two independent non-crystalline components in the smectic phase of PTFE, the first of which includes partially ordered PTFE macromolecules, and the second, disordered layers of specific structural formations, so-called hexagons [44]. This agrees with the conclusion made in [50], where the nature of halo II was attributed to the presence of low molecular weight impurities in the polymer. Possibly, the origin of halo II can be attributed to the presence of pores also (more accurately, to the presence of a smectic phase on the surface of pore walls) [51].

Using the Debye–Scherrer equation, the size of scattering centers providing the appearance of the first and second components of halo II on the diffraction pattern of PTFE was estimated to be equal 10.06 and 5.65 Å for the initial polymer and 9.69 and 5.16 Å for TRM-PTFE, respectively. As a whole, it can be concluded that the combined action of γ-radiation and high temperatures provides no noticeable changes in the characteristics of the halo II components in the X-ray spectra of the initial polymer and TRM-PTFE.

The nature of halo III in the range of diffraction angles 72–75°, observed in the X-ray spectrum of PTFE, is attributed to scattering by polymer macromolecules with a 15/7 helix conformation [46]. Unfortunately, a high signal-to-noise ratio characteristic of this region of the recorded X-ray spectra of PTFE does not allow certain conclusions about the effect of thermal-radiation modification on the characteristics of halo III; at least, it was assumed that they also change insignificantly (Table 3).

### 3.3. Determination of the Lattice Parameters for Initial and Thermal-Radiation Modified Polytetrafluoroethylene by Means of Hull-Davey Chart

In the hexagonal structure of PTFE crystals, the interplanar spacings *d_hkl_* are related to the lattice constants *a* = *b* and *c* as [40,41,42]:(1)$$\frac{1}{d_{hkl}^{2}}=\frac{4}{3}\cdot \frac{h^{2}+hk+k^{2}}{a^{2}}+\frac{l^{2}}{c^{2}}$$

Let us transform Equation (1) in accordance with [41,42] to determine the parameters of the unit cell of the initial polymer and TRM-PTFE:(2)$$2\log\left(d_{hkl}\right)=2\log a-\log\left[\frac{4}{3}\left(h^{2}+hk+k^{2}\right)+\frac{l^{2}}{{\left(c/a\right)}^{2}}\right]$$

Next, we write Equation (2) for two reflection planes of a hexagonal PTFE crystal (0015 and 1015, Table 1), and subtract one equation from the other:(3)$$2\log d_{0015}-2\log d_{1015}=-\log\left[\frac{4}{3}\left(h_{1}^{2}+h_{1}k_{1}+k_{1}^{2}\right)+\frac{l_{1}^{2}}{{\left(c/a\right)}^{2}}\right]\cdots +\log\left[\frac{4}{3}\left(h_{2}^{2}+h_{2}k_{2}+k_{2}^{2}\right)+\frac{l_{2}^{2}}{{\left(c/a\right)}^{2}}\right]=-F_{1}+F_{2}$$

Here, the difference between interplanar spacings for reflections 0015 and 1015 does not depend on the lattice constant *a*. The graphical solution of Equation (3) is the so-called Hull-Davey diagram, which is a log-lin plot of the parameter *F* = 4(*h*^2^ + *hk* + *k*^2^)/3 + *l*^2^/(*c/a*)^2^ as a function of the ratio *c/a* (Figure 5) [41,42]. Then, the point of intersection of the straight lines corresponding to the difference Δ = 2log(*d*_0015_) − 2log(*d*_1015_) for the initial polymer and TRM-PTFE with the dependence describing the difference *F*_2_ − *F*_1_ on *c/a* will determine the desired ratio of the lattice constants of PTFE (Figure 5, curves 3–5, Table 2).

The lattice constant *a* of PTFE crystals was determined using the equation
(4)$$\frac{a^{2}}{d_{hkl}^{2}}=\frac{4}{3}\cdot \left(h^{2}+hk+k^{2}\right)+\frac{l^{2}}{{\left(c/a\right)}^{2}}$$

The lattice constant a can be determined from the slope of the dependence Y = [(4/3) × (*h*^2^ + *hk* + *k*^2^) + *l*^2^/(*c/a*)^2^] ¤ 1/*d_hkl_*^2^ (Figure 6). It was found that the thermal-radiation modification of PTFE leads to an insignificant increase in the lattice constant *a* by ~0.1 Å and to an insignificant decrease in the large lattice constant *c* also by ~0.1 Å (Table 2).

### 3.4. Comparative Analysis of the Experimentally Established as Well as Obtained from X-ray Data Density Values of the Initial and Thermal-Radiation Modified Polytetrafluoroethylene 

The volume of the unit cell and the X-ray density of the initial polymer and TRM-PTFE were calculated using the following expressions [40]:(5)$$V=0.866a^{2}c$$
(6)$$\rho_{c}=\frac{M}{N_{A}V}$$
where *M* = 0.75 kg/mol is the molar mass of PTFE [46] and *N_A_* is the Avogadro number. As follows from Table 2, the volume of the unit cell in TRM-PTFE increases, which can be due to an increase in interplanar spacings in the crystalline phase of PTFE after gamma-radiation modification at high temperatures (Table 1). An increase in the volume of the unit cell in TRM-PTFE is accompanied by a decrease in *ρ_c_*, despite an increase in the degree of crystallinity of PTFE (Table 2).

Along with an insignificant increase in the degree of crystallinity in TRM-PTFE, a decrease in the average crystallite size *L* by a factor of 3.82 relative to initial PTFE is observed (Table 2). A decrease in *L* can be attributed to both the destruction of PTFE crystals during irradiation [54] as well as to crosslinking processes, which can restrict the polymer crystal growth during cooling of the melt [55].

The value of *ρ_c_* for PTFE given in Table 2 is in good agreement with the previously obtained values 2.302 g/cm^3^ [56] and 2.300 g/cm^3^ [57]. It should be emphasized that the combined action of high temperatures and γ-radiation leads to a decrease in the density of the modified polymer relative to PTFE.

Apparently, the value *ρ_c_* = 2.69 g/cm^3^ for PTFE, calculated in [46], is overestimated. This may be due to the error in determining 2θ_max_ = 16.28° for the diffraction maximum *hkl* = 003 used to calculate the large lattice constant of PTFE *c* = 16.8 Å, which further led to the underestimated volume of the PTFE unit cell (~ 470 Å^3^), in comparison with the data given in Table 2, and, further, to the overestimated degrees of crystallinity of the polymer.

However, the large lattice constant *c* value established in [46] is in good agreement with the value of this constant for PTFE macromolecules with the triclinic conformation (*a* = 5.59 Å, *c* = 16.88 Å, γ = 119.3°) [58]. Then, it can be assumed that the crystalline phase of PTFE contains crystals in both the hexagonal and triclinic conformations.

It should be noted that the presence of crystals in the triclinic conformation in the crystalline phase of PTFE, which determine its composition in the temperature range below a phase transition temperature of 19 °C, can significantly increase the density of PTFE [58]. At the same time, when studying the structure of composite materials based on TRM-PTFE filled with silicon dioxide particles, it was shown that the phase transitions in PTFE are “smoothed” [59]. Consequently, the crystalline phase in PTFE at room temperature can consist of both “triclinic” and “hexagonal” crystals, which possibly form observed two components of the diffraction maximum *hkl* = 100 in the X-ray spectra of PTFE. Therefore, it can be assumed that the previously obtained estimates of the large lattice constant c and the X-ray density of PTFE confirm this fact [46].

Thus, the thermal-radiation treatment of PTFE can lead to a decrease in the content of triclinic crystals and an increase in the content of hexagonal crystals in the crystalline phase of the polymer. Then, the X-ray density of TRM-PTFE should be lower than *ρ_c_* of PTFE, which was confirmed experimentally (Table 2).

Using the known estimates of the density *ρ_a_* of the amorphous phase in PTFE (2.056 and 2.040 g/cm^3^ in [55,56], respectively), the density *ρ* of the initial polymer and TRM-PTFE can be calculated using the relationship
(7)$$\rho_{i}=\rho_{ci}X_{ci}+\rho_{ai}\left(1-X_{ci}\right)$$
where *i* = 1, 2 for the initial polymer and TRM-PTFE, respectively. The calculations were performed with *ρ*_*a*1_ = *ρ*_*a*2_ = 2.05 g/cm^3^. As follows from Table 2, the calculated values of the polymer density are in satisfactory agreement with the experimentally obtained data on the density of the initial polymer and TRM-PTFE. The somewhat higher value of the theoretical estimate of the PTFE density in comparison with the experimentally obtained one may be due to the presence of the closed pores in the bulk of the sample.

## 4. Discussion

In conclusion, let us discuss the nature of the effect of improving the deformation-strength and tribological properties, as well as increasing the radiation resistance of PTFE, caused by thermal radiation treatment [1,2,3,4,5,6]. It is easy to estimate the density of cross-links in the polymer from the average size of the TRM-PTFE crystals ~1.32 × 10^18^ cm^−3^ (Table 2) [10]. However, in the case of acetylene-sensitized radiation crosslinking of PTFE, it was shown that noticeable changes in PTFE physical and mechanical properties begin at a crosslink density of ~10^21^ cm^−3^ [60]. Therefore, the assumption [1,2,3,4,5,6] about the radiation crosslinking of polymer chains during high-temperature radiolysis of PTFE, as the main reason for improving the operational properties of the material, is doubtful.

As follows from the results of this work, the combined action of γ-radiation and high temperatures leads to the disappearance of pores and the ordering of the crystalline phase of PTFE due to the removal of defective crystals, which may have the triclinic conformation. In this case, the degree of crystallinity increases and the size of PTFE crystals decreases, which can be significant for improving the deformation-strength and tribological characteristics of the polymer.

The conclusion about the high radiation resistance of TRM-PTFE [1,2,3,4,5,6] is erroneous, since the radiolysis of the unmodified polymer proceeds in a heterogeneous mode [61], due to the presence of the pore system found in this study. The dissolution and coalescence of pores in the course of thermal-radiation modification of PTFE leads to the implementation of a homogeneous regime of radiolysis of TRM-PTFE during repeated irradiation, and an increase in the radiation resistance of the polymer in terms of mechanical properties. Obviously, the comparison of the radiation resistance of PTFE, performed disregarding the mode, in which the polymer radiolysis proceeds, is incorrect [1,2,3,4,5,6].

In addition, the dependence of the strength characteristics of solids on the porosity is known [62]. Therefore, the removal of pores from the bulk of the polymer makes it possible to immediately improve the deformation and strength characteristics of TRM-PTFE.

Finally, let us discuss the question of the contribution of temperature and radiation factors to the thermal-radiation-induced changes in the PTFE structure discovered in this study. The combined action of ionizing radiation and high temperatures in our case leads to the realization of two effects: (i) ordering the crystal structure of polytetrafluoroethylene (PTFE) as a result of a decrease in the content of crystals with defects; (ii) the disappearance of the system of micropores formed during the preparation of the polymer sample. In addition, the influence of ionic particles in the course of radiation crosslinking of PTFE in the melt cannot be ruled out. Let us take a closer look at these effects:

(i) The mechanisms of reactions proceeding under the influence of ionizing radiation and reactions excited thermally are different [63]. The formation of final and intermediate substances under action of ionizing radiation can occur as a result of processes such as radical or ion-molecular reactions. The discussed structural changes in PTFE are a consequence of the course of radiation-induced processes of destruction and cross-linking of macromolecules [64]. In this case, the radiation destruction of the polymer differs from the thermal one, which occurs at higher temperatures. During the radiation destruction of PTFE in the melt, depolymerization and intense release of gaseous products of radiolysis are observed [9].

There are different points of view on the course of changes in the microstructure of PTFE during high-temperature radiation treatment for example, the decisive role of crosslinking processes is discussed in [65]. However, at the same time there are arguments in favor of amorphization and the accumulation of defects during irradiation of PTFE [66].

In particular, a decrease in the absorbed dose corresponding to the onset of the PTFE decrystallization process from 3 MGy to 1 MGy observed with an increase in the temperature of gamma irradiation from 303 to 453 K may be due to acceleration of the depolymerization of the polymer at high temperatures and the enhancement of chain branching [66].

The results obtained in this article are consistent with the second hypothesis. However, in addition to conclusions [66], it should be noted that at the initial stages of PTFE radiolysis (lower than 170 kGy) both at room temperature and in the temperature range from 327 to 350 °C, the ordering of its structure occurs simultaneously with an increase in the crystallinity of the polymer. An increase in the irradiation temperature has a beneficial effect on this process, contributing to the removal of defective PTFE crystals. Apparently, in the case of high-temperature irradiation the processes of radiation crosslinking primarily affect the growth of PTFE crystals, contributing to the creation of a fine-crystalline polymer structure, which is optimal for improving the deformation-strength and tribological characteristics of the material.

(ii) Radiation-induced formation of a system of closed pores is one of the main radiation effects that leads to a change in the structure of condensed media [17,18,19,67,68,69]. Discussion of possible mechanisms of pore formation during irradiation of condensed media showed that radiation-chemical processes of destruction, crosslinking and gas release have a decisive effect on the formation of a porous structure in irradiated solids [67,68]. At the same time, the radiation heating of solids, which occurs as a result of the dissipation of the absorbed energy of ionizing radiation [63,68], turns out to be insufficient for the development of pore formation processes.

This is confirmed by the results of a study by the method of X-ray computer microtomography of radiation-induced changes in the characteristics of the porous structure in foamed polyurethane samples, gamma-irradiated to various doses at room temperature [69]. It is shown that the observed changes in the porous structure of the polymer are caused by the processes of increasing the size of micropores and coalescence, which are caused by radiation crosslinking of macromolecules.

At first glance, the results [67,68,69] contradict the effect of pore dissolution discovered in our study in the case of PTFE irradiation in the melt. However, it was shown [70] that an increase in the irradiation temperature has a strong effect on the volume concentration, shape, and size of helium bubbles in boron carbide. In the region of low irradiation temperatures, the bubbles were homogeneously distributed over the volume of the sample; however, at high temperatures, they were localized near dislocations, interfaces, etc. In this case, the concentration of bubbles decreased from 10^15^–10^18^ to 10^12^ cm^−3^. It was concluded that with an increase in the irradiation temperature, the mechanism of pore formation changes from homogeneous to heterogeneous. It was also noted [71] that the processes of pore coalescence can occur only under high-temperature irradiation, when the irradiation temperature exceeds half the melting temperature of the material under study.

The data obtained in this study on the regularities of the behavior of the porous structure of PTFE under the conditions of polymer irradiation in the melt are in full agreement with the conclusions [70,71]. Thus, the temperature factor has a decisive effect on the behavior of the porous structure of PTFE under irradiation.

(iii) It is assumed in studies [1,2,3,4,5,6] that the crosslinking reaction of polymer chains under high-temperature irradiation of PTFE proceeds by a radical mechanism. At the same time, the contribution from possible reactions with the participation of ionic particles is not taken into account. Indeed [63], stabilization and accumulation of significant concentrations of ions and ion-radicals should be expected upon low-temperature irradiation of PTFE. Nevertheless, the detection of radiation-induced hole conductivity of PTFE irradiated at room and elevated temperatures suggests that a certain steady-state concentration of “parent” cations can be established in the PTFE melt during high-temperature irradiation [72,73]. These cations can react with macromolecules and form crosslinks [74]. If the temperature dependence of radiation-chemical processes with the participation of radicals obeys the Arrhenius law, then the regularities of the change in the rate of ion-molecular reactions with increasing temperature are still unknown [63,74]. Therefore, it is not possible to estimate the contribution of radiation-induced processes and temperature to the development of processes of radiation-chemical crosslinking of PTFE.

Apparently, it is impossible to describe the contribution of radiation and temperature factors to the development of radiation-induced processes in PTFE within the framework of one theoretical model. The above discussion of the effect of temperature on various radiation-induced processes developing in the PTFE melt during radiation modification shows that for each type of such processes it is necessary to take into account the contribution of temperature and radiation-chemical reactions separately. Obviously, it is necessary to carry out further studies of the regularities of changes in the structure and radiation-chemical processes in PTFE depending on the irradiation temperature.

## 5. Conclusions

The study of changes in the microstructure and structure of the crystalline phase of PTFE caused by the combined action of γ-radiation and high temperatures has provided the following conclusions:

(i) The existence of a developed porous structure with an average pore diameter of ~3.5 μm and a pore volume content of ~1.1% in PTFE has been revealed by the technique of X-ray computed microtomography. The thermal-radiation modification of PTFE leads to the disappearance of the porous structure and the formation of several large pores with sizes from 30 to 50 μm.

(ii) A comparative study of the diffraction patterns of the initial polymer and TRM-PTFE has been carried out by the method of X-ray diffraction analysis. It has been established that the combined action of γ-radiation up to a dose of 173 kGy and high temperatures leads to an increase in interplanar spacings, the degree of crystallinity, and volume of the unit cell and to a decrease in the size of crystals and the X-ray density of the crystalline phase in TRM-PTFE in comparison with the unmodified polymer. All found differences have been explained under the assumption that the phase transition in the crystalline phase of PTFE at 19 °C is smoothed; as a result, crystals of triclinic and hexagonal conformations can be present in PTFE at room temperature.

(iii) Thermal-radiation modification leads to the removal of crystals in the triclinic conformation, to an increase in the contribution of hexagonal crystals, and to local ordering of the TRM-PTFE structure. Satisfactory agreement between the calculated and experimentally determined values of the density of the initial polymer and TRM-PTFE confirms the assumption about the presence of two types of lattices in the crystalline phase of PTFE

(iv) The effect of improving the deformation-strength and tribological properties of PTFE caused by the combined action of ionizing radiation and high temperatures is due not only to the processes of radiation cross-linking of polymer chains but also to the disappearance of the pore system and the ordering of the crystalline phase of PTFE. Radiation crosslinking processes can affect the growth and size of the resulting PTFE crystals.

(v) Averaging of the properties of PTFE under the combined action of the high temperatures and gamma-irradiation is the main reason for a significant increase in the radiation resistance of the thermal-radiation-modified polymer.

## Figures and Tables

**Figure 1 polymers-13-03678-f001:**
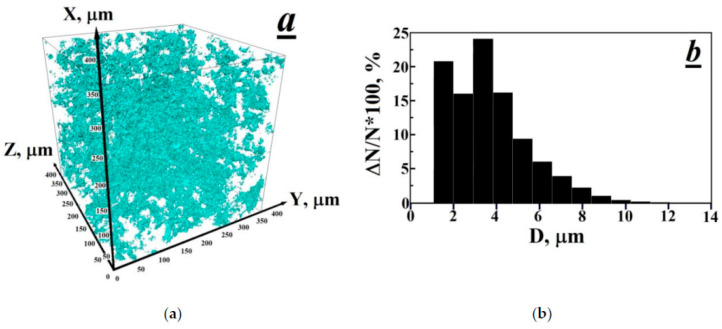
Three-dimensional image X-ray computed microtomography of a porous structure in the volume of a polytetrafluoroethylene sample (**a**) and the histogram of the size distribution of pores (**b**).

**Figure 2 polymers-13-03678-f002:**
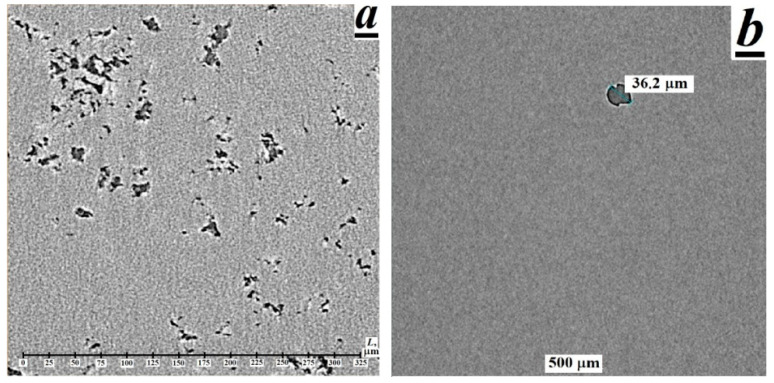
Electronic images of the microstructure of the (**a**) initial and (**b**) thermal-radiation-modified polytetrafluoroethylene samples, recorded by X-ray computed microtomography by scanning the samples at the half of the height of the sample.

**Figure 3 polymers-13-03678-f003:**
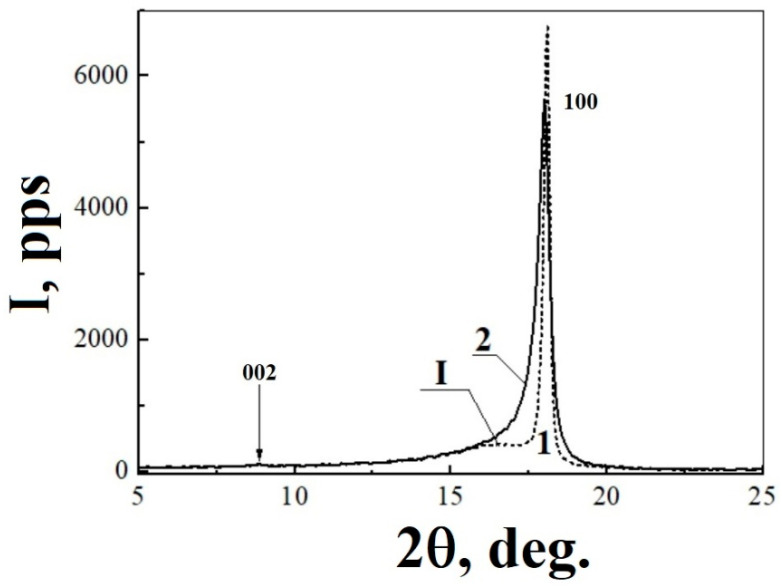
Diffraction patterns of the (1) initial and (2) thermal-radiation-modified polytetrafluoroethylene samples in the range of diffraction angles 2θ = 5–25°: (I) is the diffuse diffraction maximum at 2θ = 10–25° and 002 and 100 are the meridional and equatorial reflections of polytetrafluoroethylene, respectively.

**Figure 4 polymers-13-03678-f004:**
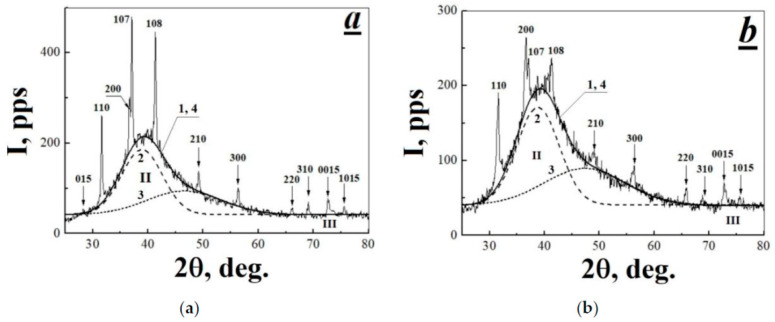
Diffraction patterns of the (**a**) initial and (**b**) thermal-radiation-modified polytetrafluoroethylene samples in the range of diffraction angles 2θ = 5–25°: (II) and (III) are the diffuse diffraction maxima in ranges of 25–70° and 72–75°, respectively; 1 is the experimental dependence; 2 and 3 are the first and second components of the 25–70° diffuse diffraction maximum established by the nonlinear least-squares method, respectively; and 4 is the approximation. Arrows and numbers indicate the diffraction maxima found in the X-ray scattering spectra of the initial and thermal-radiation-modified polytetrafluoroethylene.

**Figure 5 polymers-13-03678-f005:**
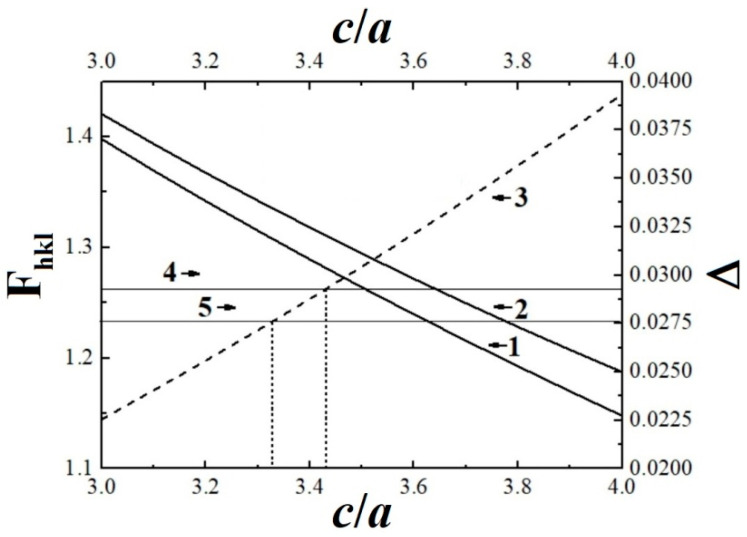
Log-lin plots (Hull–Davey diagrams) of (1, 2) the parameter *F* = 4(*h*^2^ + *hk* + *k*^2^)/3 + *l*^2^/(*c/a*)^2^ for reflections *hkl* = (1) 0015 and (2) 1015 in the X-ray spectrum of polytetrafluoroethylene, (3) the difference F_1015_–F_0015_, and (4, 5) Δ = 2log (d_0015_) − 2log (d_1015_) for the (4) initial and (5) thermal-radiation-modified polytetrafluoroethylene samples versus the ratio of the hexagonal lattice constants *c/a*. Here, *d*_0015_ and *d*_1015_ are the interplanar spacings for reflections *hkl* = 0015 and 1015, respectively. The left and lower axes refer to plots 1–3, whereas the right and upper axes refer to plots 4 and 5. The vertical dashed lines mark the points of intersection of straight lines 4 and 5 with dependence 3.

**Figure 6 polymers-13-03678-f006:**
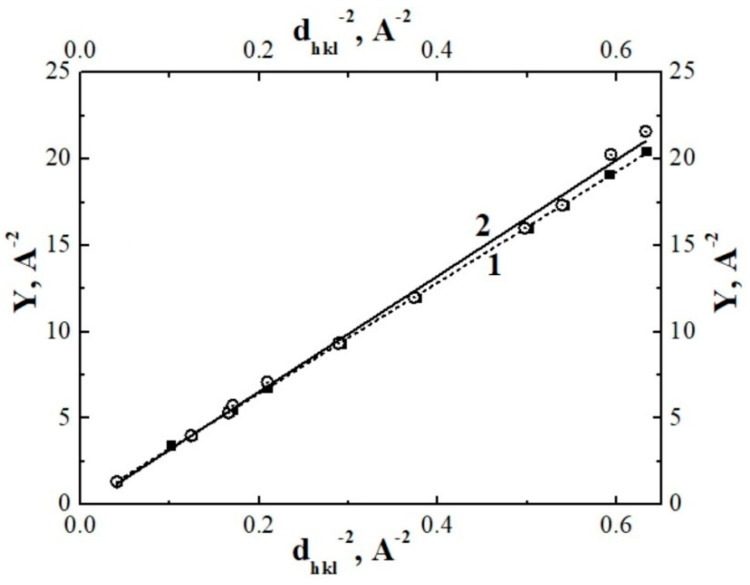
Determination of the crystal cell parameter *a* in samples of initial and thermo-radiation-modified polytetrafluoroethylene (dependences 1, 2, respectively). Linear dependences 1 and 2 are obtained by the least-squares method.

**Table 1 polymers-13-03678-t001:** Characteristics of the X-ray scattering spectra of the initial and thermal-radiation-modified polytetrafluoroethylene samples.

Miller Indexes			Initial Polytetrafluoroethylene				Thermal-Radiation-Modified Polytetrafluoroethylene			
*h*	*k*	*l*	2θ_max_, deg.	*I*, pps	*I*/*I*_max_, %	*d_hkl_*, Å	2θ, deg.	*I*, pps	*I*/*I*_max_, %	*d_hkl_*, Å
0	0	2	8.848	17.637	0.283	9.986	8.803	40.79	0.727	10.037
1	0	0	18.076	6229.521	100	4.904	17.955	5607.94	100	4.936
0	1	5	28.330	10.889	0.175	3.148	-	-	-	-
1	1	0	31.619	178.654	2.868	2.827	31.515	110.45	1.970	2.837
2	0	0	36.655	123.719	1.986	2.450	36.569	100.18	1.786	2.455
1	0	7	37.091	279.397	4.485	2.422	37.110	70.18	1.251	2.421
1	0	8	41.345	227.647	3.654	2.182	41.286	51.29	0.915	2.185
2	1	0	49.208	40.559	0.651	1.850	48.932	26.23	0.468	1.860
3	0	0	56.348	39.484	0.634	1.631	56.175	25.69	0.458	1.636
2	2	0	66.131	15.177	0.244	1.412	65.820	20.51	0.366	1.418
3	1	0	69.080	24.203	0.389	1.359	68.887	31.84	0.568	1.362
0	0	15	72.693	33.577	0.539	1.230	72.821	15.43	0.275	1.298
1	0	15	75.614	16.701	0.268	1.257	75.572	9.28	0.165	1.257

Note: 2θ_max_, deg. is the position of the maximum of the diffraction peak in the X-ray spectrum; *I*, pps (pulse per second) is the intensity of the diffraction maximum; *I*/*I*_max_, % is the relative intensity of the diffraction maxima in the X-ray spectrum.

**Table 2 polymers-13-03678-t002:** Degree of crystallinity *X_c_*, average crystallite size *L*, lattice constants *a* and *c*, volume *V* of the unit cell, and the X-ray density *ρ_c_* of the initial and thermal-radiation-modified polytetrafluoroethylene samples.

Description of Characteristics, Dimension	Initial Polytetrafluoroethylene	Thermal-Radiation-Modified Polytetrafluoroethylene
*X_c_*, %	65.04	77.02
*L*, Å	348.588	91.138
*c*/*a* *	3.432	3.329
*a*, Å	5.664 ± 0.377	5.792 ± 0.784
*c*, Å	19.429 ± 1.293	19.299 ± 2.612
*V*, Å^3^	539.778	560.673
*ρ_c_*, g/cm^3^	2.307	2.221
*ρ*_theor_, g/cm^3^	2.217	2.182
*ρ*_exp_, g/cm^3^	2.205	2.180

Note: (*)-determined graphically from the Hull–Davey chart (Figure 5).

**Table 3 polymers-13-03678-t003:** Diffuse diffraction maxima in the X-ray scattering spectrum of the initial and thermal-radiation-modified polytetrafluoroethylene samples and its characteristics.

No.	Δ(2θ)_halo_, deg.	2θ_max_, deg.	A, deg.·pps	*b*, deg.	*H*, pps
Initial polytetrafluoroethylene					
I	10–25	16.670 ± 0.059	1546.042 ± 62.703	3.729	389.496
IIa	25–70	38.905 ± 0.054	1356.051 ± 115.925	8.747	145.638
IIб	25–70	46.603 ± 1.003	894.570 ± 131.306	15.989	52.561
III	72–75	73.418 ± 0.105	10.020 ± 2.498	0.914	10.304
Thermal-radiation-modified polytetrafluoroethylene					
I	10–25	16.680 ± 0.043	1626.748 ± 45.961	3.404	448.911
IIa	25–70	38.807 ± 0.045	1258.245 ± 81.325	9.075	130.246
IIб	25–70	47.350 ± 0.778	919.392 ± 96.847	17.545	49.229
III	72–75	74.135 ± 0.145	9.001 ± 2.368	1.408	6.007

Note: Δ(2θ)_halo_ is the diffraction angle range in which the diffuse diffraction maximum is observed in polytetrafluoroethylene X-ray spectra, A is the integral intensity, 2θ_max_ is the position of the maximum of the diffraction peak in the X-ray spectrum, *b* is the FWHM of the diffuse diffraction maximum, and *H* is the amplitude of the diffuse diffraction maximum.

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
