# Peer review of "Structure of Polytetrafluoroethylene Modified by the Combined Action of γ-Radiation and High Temperatures"

_polymers, 2021, doi:10.3390/polym13213678_

Round 1
Reviewer 1 Report
The manuscript presents the results of an investigation of the polytetrafluoroethylene (PTFE) modified by gamma-radiation and high temperature. It reports one X-ray microtomography result, one electronic image, diffraction patterns and a calculation of lattice pattern.
PTFE is a challenging material to work with. One would like to see a statistical analysis of the porosity (over 10 or so samples, for example).
It is not clear how the electronic image was been obtained: technical details of the instrument and coating protocol are not given.
The pattern analysis seems to be accurate, but only the crystallography community can judge this work.
Hence, I advise to resubmit this work to another journal.
Author Response
REPLY TO REVIEW
Question 1: Extensive editing of English language and style required
Answer: The authors checked the English language and style of the text again and made the necessary changes.
| Existing text | Corrected text |
| Page 3, paragraph 1: | |
| At the same time, it is known that traces of the crystalline phase of PTFE can exist in the melt up to a temperature of 400 °C and that the processes of radiative gas evolution and depolymerization under γ-irradiation of PTFE up to 500 °C proceed [8, 9]. | At the same time, it is known that traces of the crystalline phase of PTFE can exist in the melt up to a temperature of 400 °C and that the processes of radiolytic gas evolution and depolymerization under γ-irradiation of PTFE up to 500 °C proceed [8, 9]. |
| Page 6-7, paragraph 6(1): | |
| The diffraction patterns of the initial and TRM-PTFE were indexed by the method described in [14–16], taking into account the reference data (protocol JCPDS No. 47-2217 [17]) and the results previously obtained in [18–20]. The procedure for determining the degree of crystallinity Xc of polymers was described earlier [11]. The interplanar spacing dhkl and the average crystal size L in the initial polymer and TRM-PTFE were calculated using the well-known Wulff–Bragg condition and Debye–Scherrer equation (Tables 1, 2) [14]. | The diffraction patterns of the initial and TRM-PTFE were indexed by the method described in [14–16], taking into account the reference data (protocol JCPDS No. 47-2217 [17]) and the results previously obtained in [18–20]. The procedure for determining the degree of crystallinity Xc of polymers was described earlier [11]. The interplanar spacing dhkl and the average crystal size L in the initial polymer and TRM-PTFE were calculated using the well-known Bragg’s law and Debye–Scherrer equation (Tables 1, 2) [14]. |
Question 2: The manuscript presents the results of an investigation of the polytetrafluoroethylene (PTFE) modified by gamma-radiation and high temperature. It reports one X-ray microtomography result, one electronic image, diffraction patterns and a calculation of lattice pattern.
Answer: The authors disagree with the remark of the reviewer, who claims that the manuscript contains only one result of the study of polytetrafluoroethylene (PTFE). In fact, the article provides a comparative study of structural changes in various materials: PTFE and thermo-radiation-modified polytetrafluoroethylene (TRM-PTFE).
The main attention was paid to the study of the regions of the TPM-PTFE sample, which contained micropores, and to the comparison of the characteristics of the pore structure in PTFE and TPM-PTFE (Fig. 2, a, b).
An analysis of two diffraction patterns of PTFE and TRM-PTFE samples and the calculation of their lattice parameters are also presented.
Question 3: PTFE is a challenging material to work with. One would like to see a statistical analysis of the porosity (over 10 or so samples, for example).
Answer: The authors, of course, agree with the reviewer that PTFE, like other polymeric materials, is among the objects of research that are difficult to study. At the same time, if we want to make sure that porosity changes have occurred in the material modified by the combined action of γ-radiation and high temperatures, there is no need to carry out a statistical analysis (it is only necessary if the changes are very small and the measurement methods are insensitive). In this case, the changes in porosity are quite large, and a new highly sensitive method of non-destructive control of the structure of materials - X-ray computer microtomography (ХCMT) - was used to analyze the porosity.
At the same time, we understand that the observed disappearance of the porous structure of PTFE as a result of thermal radiation treatment deserves further study, which will be done in the next work. For a better understanding of the importance of using the RCMT method for studying porosity, a brief overview of methods for determining porosity has been added to the "Experimental Technique" section of the article, and a discussion of the need to study radiation-induced changes in the porous structure of PTFE depending on the irradiation temperature has been added in the "Discussion" section.
Question 4: It is not clear how the electronic image was been obtained: technical details of the instrument and coating protocol are not given.
Answer: The technical characteristics of the FEI HELISCAN MICRO-CT microtomograph used in the research can be found on the website of the manufacturer - TermoFisher Scientific:
https://www.thermofisher.com/ru/ru/home/electron-microscopy/products/ microct/heliscan-microct.html
The specified link, as well as a detailed description of the procedure for performing measurements by the XCMT method and obtaining electronic images were added to the text of the article in the section "Experimental Technique".
Question 5: The pattern analysis seems to be accurate, but only the crystallography community can judge this work.
Answer: One of the authors of the article is Professor E.D. Politova, who is a specialist in the field of X-ray phase analysis of the structure of materials [1-3], and a member of the crystallographic community mentioned by the reviewer.
References
1. Tian Ye, Jing Li, Qingyuan Hu, Jin Li, Kun Yu, Jinglei Li, Politova E.D., Stefanovich S.Yu, Zhuo Xu, Xiaoyong Wei // Ferroelectric Transitions in Silver Niobate Ceramics // Journal of Materials Chemistry С. – 2019. – V. 7. – pp. 1028 – 1034 doi:10.1039/C8TC05451G
2. Politova E., Golubko N., Kaleva G., Mosunov A., Sadovskaya N., Stefanovich S., Panda P. // Structure, dielectric and ferroelectric properties of perovskite ceramics // Acta Crystallographica Section A: Foundations and Advances. – 2017. – No. A2. – pp. C990 doi: 10.1107/S2053273317085849
3. Ivanov S.A., Nordblad P., Mathieu R., Tellgrence R., Ritter C., Politova E., Kaleva G., Mosunov A., Stefanovich S., Weil M. // Spin and Dipole Ordering in Ni2InSbO6 and Ni2ScSbO6 with corundum-related structure // Chemistry of Materials. – 2013. – V.25, No. 3. – pp. 935 – 945 doi: 10.1021/cm304095s
Question 6: Hence, I advise to resubmit this work to another journal.
Answer: The authors believe that their answers will be convincing to the editorial board and the reviewer. At the same time, we are grateful to the referee, since his comments were useful, and taking them into account allowed us to improve the text of the article.

Reviewer 2 Report
In this manuscript, the authors report on structure of polytetrafluoroethylene modified by the combined action of γ-radiation and high temperatures. The manuscript reports interesting and important results in the field, therefore my recommendation is to accept it for publication in Polymers Journal, subject to the following minor revision point:
It will be interesting if authors could provide a more discussion regarding the competition between the influence of heating and gamma-irradiating on the structure changes of polytetrafluoroethylene. Is it possible to quantify separately these two influences?
Author Response
REPLY TO REVIEW
Question: It will be interesting if authors could provide a more discussion regarding the competition between the influence of heating and gamma-irradiating on the structure changes of polytetrafluoroethylene. Is it possible to quantify separately these two influences?
Answer: The combined action of ionizing radiation and high temperatures in our case leads to the realization of two effects: i) ordering the crystal structure of polytetrafluoroethylene (PTFE) as a result of a decrease in the content of crystals with defects; ii) the disappearance of the system of micropores formed during the preparation of the polymer sample. Let's take a closer look at these effects:
i) The mechanisms of reactions proceeding under the influence of ionizing radiation and reactions excited thermally are different [1]. The formation of final and intermediate substances can occur as a result of processes such as radical reactions or ion-molecular reactions. The discussed structural changes in PTFE are a consequence of the course of radiation-induced processes of destruction and cross-linking of macromolecules [2]. In this case, the radiation destruction of the polymer differs from the thermal one, which occurs at higher temperatures. During the radiation destruction of PTFE in the melt, depolymerization and intense release of gaseous products of radiolysis are observed [3]. An increase in the PTFE irradiation temperature leads to an increase in the radiation-chemical yield of destruction in the amorphous phase of the polymer and, accordingly, to an increase in the degree of crystallinity [4]. In the case of irradiation of PTFE at a temperature of 303 K, a sharp decrease in the degree of crystallinity of PTFE occurs in the dose range from 3 to 10 MGy. An increase in the irradiation temperature to 453 K makes it possible to reduce the dose at which the degree of crystallinity of the polymer begins to decrease, to 1 - 3 MGy. The origin of this radiation effect can be associated with the intensification of the depolymerization processes, which leads to the amorphization of the polymer and the enhancement of chain branching in PTFE.
There are different points of view on the course of changes in the microstructure of PTFE during high-temperature radiation treatment: in [5], the decisive role of crosslinking processes is discussed, at the same time there are arguments in favor of amorphization of PTFE and the accumulation of defects during irradiation [4].
Such conclusions can be drawn from the currently available experimental material [1-5]. Further research is needed for a more substantive discussion of the competition between heating and gamma irradiation in structural changes in polytetrafluoroethylene. The above considerations have been added to the article in the "Discussion" section.
ii) Radiation-induced formation of a system of micropores is one of the main radiation effects leading to a change in the structure of condensed media [6 - 9]. Discussion of possible mechanisms of micropore formation during irradiation of condensed media showed that radiation-chemical processes of destruction and crosslinking, gas release have a decisive effect on the formation of a porous structure in irradiated solids [6, 9]. At the same time, the radiation heating of solids, which occurs as a result of the dissipation of the absorbed energy of ionizing radiation [1, 9], turns out to be insufficient for the development of pore formation processes.
This is evidenced by the results of a study by the method of X-ray computer microtomography of radiation-induced changes in the characteristics of the porous structure in foamed polyurethane samples, gamma-irradiated to various doses at room temperature [8]. It is shown that the observed changes in the porous structure of the polymer are caused by the processes of increasing the size of micropores and coalescence, which are caused by radiation crosslinking of macromolecules.
However, the conclusions [8] contradict the results obtained in our article. The reason for the contradiction may be related to the effect of temperature: probably, during irradiation in the melt, the processes of dissolution and coalescence of micropores proceed faster in comparison with the above-mentioned set of radiation-chemical processes, providing the appearance of pores in PTFE.
Therefore, the assessment of the contribution of temperature and radiation factors during the high-temperature radiolysis of PTFE should be carried out separately for both the processes of radiation-induced changes in the porous structure of the polymer and the processes developing in the crystalline phase of PTFE. These circumstances do not allow one to quantify the contributions to the change in the PTFE structure associated with the action of ionizing radiation and temperature.
References
1. Swallow A.J. // Radiation Chemistry of Organic Compounds. – Oxford London New-York Paris: Pergamon Press, 1960. - 380 p.
2. Wojna´rovits L. // 23 Radiation Chemistry // in: Handbook of Nuclear Chemistry / Attila Ve´rtes, Sa´ndor Nagy, Zolta´n Klencsa´r, Rezso˝ G. Lovas, Frank Ro¨sch (Eds.). - Springer Science+Business Media B.V., 2011. – pp. 1267 - 1331 doi:10.1007/978-1-4419-0720-2
3. E. Roland, R.E. Florin, M.S. Parker, L.A. Wall // The Mechanism of the Depolymerization of Polytetrafluoroethylene With Pyrolytic and Radiolytic Initiation // J. Res. Nat. Bur. Stand. A. Phys. Chem. – 1966 – V. 70. – pp. 115 – 131 doi: 10.6028/jres.070A.008
Briskman B.A., Tlebaev K.B. // Radiation Effects on Thermal Properties of Polymers. II. Polytetrafluoroethylene // High Performance Polymers. – 2008. - 20: 86. – pp. 86 – 114 doi: 10.1177/0954008307079540
Hill D.J.T., Whittaker A.K. // Radiation chemistry of polymers / in: Encyclopedia of polymer science and technology / ed. by Herman F. Mark. - John Wiley & Sons, 2016. – 58 p. doi: 10.1002/0471440264.pst488.pub2
Geguzin J.E. // Physics of Sintering. – Moscow: Nauka, 1984. – 312 p.
Sueo Machi, Silverman J. // Bubble Formation in Radiation-Induced Grafting of Styrene to Polyethylene // Journal of Polymer Science: Part A-1. – 1969. – V. 7. – pp. 2737 – 2740
Agrawal K., Singh B., Kashyap Y.S., Shukla M., Manjunath B.S., Gadkari S.C. // Gamma-irradiation-induced micro-structural variations in flame-retardant polyurethane foam using synchrotron X-ray micro-tomography // Journal of Synchrotron Radiation. - 2019. – V. 26. – pp. 1797–1807 doi: 10.1107/S1600577519009792
Grogan J.M., Schneider N.M., Ross F.M., Bau H.H. // Bubble and Pattern Formation in Liquid Induced by an Electron Beam // Nano Letters. – 2014. – V. 14. – pp. 359 - 364 doi: 10.1021/nl404169a

Round 2
Reviewer 1 Report
Unfortunately, I feel this crystallography paper should be published in a crystallography journal, not in Polymers.